# Emergence and clonal expansion of a *qacA*-harbouring sequence type 45 lineage of methicillin-resistant *Staphylococcus aureus*

Yi Nong [1] ✉, Eike Steinig[2], Georgina L. Pollock [2], George Taiaroa[2], Glen P. Carter [1,3], Ian R. Monk [1], Stanley Pang [4,5], Denise A. Daley[6], Geoffrey W. Coombs[4,5], Brian M. Forde [7], Patrick N. A. Harris [7,8], Norelle L. Sherry [1,9], Benjamin P. Howden [1,9], Shivani Pasricha[2], Sarah L. Baines [1,11] & Deborah A. Williamson [2,10,11] ✉

The past decade has seen an increase in the prevalence of sequence type (ST) 45 methicillin-resistant *Staphylococcus aureus* (MRSA), yet the underlying drivers for its emergence and spread remain unclear. To better understand the worldwide dissemination of ST45 *S. aureus*, we performed phylogenetic analyses of Australian isolates, supplemented with a global population of ST45 *S. aureus* genomes. Our analyses revealed a distinct lineage of multidrug-resistant ST45 MRSA harbouring *qacA*, predominantly found in Australia and Singapore. Bayesian inference predicted that the acquisition of *qacA* occurred in the late 1990s. *qacA* was integrated into a structurally variable region of the chromosome containing Tn*552* (carrying *blaZ*) and Tn*4001* (carrying *aac(6')-aph(2")*) transposable elements. Using mutagenesis and in vitro assays, we provide phenotypic evidence that *qacA* confers tolerance to chlorhexidine. These findings collectively suggest both antimicrobial resistance and the carriage of *qacA* may play a role in the successful establishment of ST45 MRSA.

Infections caused by *Staphylococcus aureus* are among the most common bacterial infections worldwide[1]. These infections can lead to serious complications with increased morbidity, placing a substantial burden on healthcare systems. Colonisation by *S. aureus* is a known predisposing factor in the development of infections; as such, decolonisation using antimicrobials and/or biocides may be used to reduce healthcare-associated *S. aureus* infections[2]. Quaternary ammonium compounds (QACs), notably chlorhexidine, are routinely used as decolonising agents for hand hygiene and pre-surgical antisepsis[3]. However, it is possible that the widespread use

of chlorhexidine exerts selective pressure on strains displaying tolerance to this biocide[3,4].

Bacterial tolerance to chlorhexidine can be mediated by QAC multidrug efflux systems, with QacA (encoded by a plasmid-borne *qacA* gene) being the predominant type in *S. aureus*[5]. Of particular concern is the increasing prevalence of *qacA* carriage in methicillin-resistant *S. aureus* (MRSA)[6]. This is because MRSA is a primary target for infection prevention and control practices, and *qacA* carriage and possible development of co-resistance to mupirocin or/and other antimicrobials pose a potential risk to

[1]Department of Microbiology and Immunology, The University of Melbourne at the Peter Doherty Institute for Infection and Immunity, Melbourne, VIC, Australia. [2]Department of Infectious Diseases, The University of Melbourne at the Peter Doherty Institute for Infection and Immunity, Melbourne, VIC, Australia. [3]Doherty Applied Microbial Genomics, Doherty Institute, The University of Melbourne, Melbourne, VIC, Australia. [4]Antimicrobial Resistance and Infectious Diseases Research Laboratory, Murdoch University, Murdoch, WA, Australia. [5]Department of Microbiology, PathWest Laboratory Medicine-WA, Fiona Stanley Hospital, Murdoch, WA, Australia. [6]Australian Group on Antimicrobial Resistance, Fiona Stanley Hospital, Murdoch, WA, Australia. [7]The University of Queensland, Faculty of Medicine, UQ Centre for Clinical Research, Brisbane, QLD, Australia. [8]Central Microbiology, Pathology Queensland, Royal Brisbane and Women's Hospital, Brisbane, QLD, Australia. [9]Microbiological Diagnostic Unit Public Health Laboratory, Department of Microbiology & Immunology, The University of Melbourne at The Doherty Institute for Infection and Immunity, Melbourne, VIC, Australia. [10]Victorian Infectious Diseases Reference Laboratory, Royal Melbourne Hospital at the Peter Doherty Institute for Infection and Immunity, Melbourne, VIC, Australia. [11]These authors contributed equally: Sarah L. Baines, Deborah A. Williamson. ✉e-mail: yi.nong@unimelb.edu.au; deborah.williamson@unimelb.edu.au

the success of these programmes[6]. In vitro studies have demonstrated an association between *qacA* carriage and chlorhexidine tolerance in *S. aureus*[7,8]. However, the clinical impact of low-level chlorhexidine tolerance conferred by *qacA* remains unclear, as in-use clinical concentrations are usually several orders of magnitude higher than the minimum concentrations required to inhibit bacterial growth, and chlorhexidine exposure may not always enrich for *qacA*-harbouring isolates as observed in clinical studies[9,10]. In addition, discrepancies between *qacA* carriage and phenotypic tolerance to chlorhexidine (minimum inhibitory concentration (MIC) < 4 mg/L) have been reported at the population level[11,12]. Understanding of *qacA*-mediated chlorhexidine tolerance is further complicated by the lack of a standardised method for biocide testing[13]. The presence of other drug efflux systems, e.g., those encoded by *norA* and *mepA* may partly mediate tolerance to chlorhexidine[14]. Hence, this knowledge gap highlights an urgent need for combined genomic and phenotypic analyses of biocide tolerance to help elucidate the contribution of specific genes, including *qacA*, to chlorhexidine tolerance in *S. aureus*, particularly MRSA.

While a recent study has adopted a combinatorial approach to uncovering the evolutionary benefit of *qacA*-mediated chlorhexidine tolerance in a dominant hospital-associated sequence type (ST) 239 MRSA lineage in Australia[15], the molecular epidemiology of biocide tolerance in MRSA of other STs is largely unexplored. Currently in Australia, the rate of MRSA infections across the country remains stable, yet the clonal composition of MRSA is fluid and varies between states and territories over time[16]. Noticeably over the past decade, ST45 MRSA has become a major MRSA lineage associated with both hospital and community-onset infections[16]. Recent surveillance of *S. aureus* bacteraemia in Australia indicated that ST45 MRSA accounted for 10.1% of total MRSA infections, 67.3% of which were community onset[17]. Of particular note, the increased rate of multidrug resistance among community-associated MRSA (i.e., from 9.2% in 2013 to 13.7% in 2019) was primarily driven by ST45[17]. This clonal expansion was noted in one study from New South Wales (NSW), where the prevalence of ST45 MRSA increased from 0.4 to 14% between 2012 and 2017[18]. An investigation into local outbreaks of ST45 MRSA in NSW revealed carriage of genetic determinants associated with resistance to aminoglycosides, macrolides, and tetracyclines[19]. Further, multidrug-resistant ST45 MRSA has been associated with hospital outbreaks outside Australia[20,21], with a high carriage rate of *qacA* notified in the Singaporean healthcare system[12]. The acquisition of the above antimicrobial resistance (AMR) and biocide tolerance determinants may be an important contributor to the spread of ST45 MRSA in the healthcare setting. Nonetheless, to our knowledge, the contribution of *qacA* to the expansion of ST45 MRSA in Australia has not been explored. Given the worldwide dissemination of ST45 *S. aureus*[22], this knowledge gap warrants the need to investigate the genomic correlation between *qacA* and the evolutionary dynamics of ST45 MRSA.

In this study, we conducted phylogenetic analyses using globally representative ST45 *S. aureus*, coupled with phenotypic characterisation of *qacA*-mediated biocide tolerance. Collectively, these data provide valuable information on the potential drivers for the widespread dissemination of ST45 MRSA.

## Results

### Complete genome sequence of a *qacA*-harbouring ST45 MRSA strain AUSMDU00020487

A representative *qacA*-harbouring ST45 MRSA, namely AUSMDU00020487, was selected as the reference strain for phylogenetic analyses and phenotypic characterisation in this study. This strain was obtained from a prospective state-wide surveillance study undertaken in Victoria, Australia in 2018[23]. The genome of AUSMDU00020487 consisted of a single chromosome (GenBank accession number: CP138566, 2,895,690 bases, 32.8% GC content) and one plasmid (GenBank accession number: CP138567, 34,997 bases, 29.0% GC content). The chromosome harboured *mecA* located in a staphylococcal cassette chromosome *mec* (SCC*mec*) type V element; biocide tolerance genes (*qacA, qacR*); and resistance genes to penicillins (*blaZ*, encoding PC1 β-lactamase), tetracyclines (*tetK*),

macrolides (*ermC*), and aminoglycosides (*aac(6')-aph(2")*). Additional resistance genes to macrolides (*ermA*) and tetracyclines (*tetM*) were found on the plasmid.

### Divergent global evolution of ST45 *S. aureus*

Phylogenetic analyses utilised genomic data from 210 clonal complex (CC) 45 MRSA isolates recovered from Australian surveillance and epidemiological studies[23–27] (Supplementary Note 1 & Supplementary Data 1). These sequences were supplemented with 1,503 ST45 *S. aureus* de novo genome assemblies from Staphopia[28], and publicly available sequence data from 288 ST45 *S. aureus* genomes used by Effelsberg et al.[22]. In total, the final dataset contained 2,001 ST45 *S. aureus* genomes.

To investigate the evolution of *qacA*-harbouring ST45 MRSA, core genome single nucleotide polymorphism (SNP) analysis of the above global collection highlighted an early divergence of ST45 *S. aureus* into two distinct lineages (Fig. 1a). Bayesian analysis of population structure (BAPS) identified four major clades, with the BAPS 1 and 2 clades appearing interleaved (Fig. 1a). Some geographic patterns were observed. The largest lineages (BAPS 1 and 2) comprised isolates predominantly from Europe and North America. The smaller lineage (BAPS 3) consisted exclusively of isolates from Africa. The fourth lineage (BAPS 4) mainly contained isolates from Australia and Southeast Asia. Of the entire dataset, the vast majority of *qacA*-harbouring isolates (97.4%, 411/422) belonged to SCC*mec* type V (Supplementary Data 1). Notably, the isolates in the BAPS 4 clade were predominantly MRSA (98.7%, 608/616) and accounted for 97.6% (412/422) of *qacA*-harbouring *S. aureus* in the wider dataset, although this may represent sampling bias (as outlined in the discussion). Within BAPS 4, 99.8% (411/412) of *qacA*-harbouring isolates were found in the BAPS 4.2 subclade (Fig. 1b), consisting of isolates from Australia, Singapore, and the United Kingdom (UK). Of the *qacA*-positive isolates in this subset, ~1.0% (4/412) were from the UK, 32.0% (132/412) were from Singapore, 17.0% (70/412) were from Australia, and the remaining 206 isolates had an unknown geographic origin.

### Emergence of a distinct ST45 MRSA lineage is associated with *qacA* acquisition

To gain insights into the temporal evolution of the BAPS 4 clade, 385 representative isolates with geographic information and year of collection details were selected for phylogenetic reconstruction. For population genetic and comparative genomic analyses, these isolates were divided into seven subclades, numbered C1 to C7, based on temporal phylogeny, geographic origin, and presence/absence of key antimicrobial resistance determinants including *qacA* (Fig. 2a). Noticeably, *qacA* was harboured by 99.1% (107/108), 95.8% (95/107), and 100% (3/3) of isolates in the C1 (Singaporean), C2 (Australian-Singaporean), and C3 (UK) subclades, respectively. The oldest *qacA*-harbouring isolate in the BAPS 4 clade was collected from Singapore in 2009, while our model predicted that the acquisition of *qacA* occurred in a MRSA background in the late 1990s (estimate: 1998, 95% highest posterior density [HPD] 1996–1999). *qacA* frequently co-occurred with *aac(6')-aph(2")* (85.5%, 185/219), *ermC* (71.7%, 157/219), and *tetK* (88.1%, 193/219) among isolates in the C1, C2, and C3 subclades. However, a lack of these AMR determinants including *qacA* was noted in the most recent Australian isolates (69.2%, 9/13) collected in 2019, resulting in the separation of C2A subclade for further investigation on gene loss. Of further note, SCC*mec* type differed between the *qacA*-negative (Type IV, 9/13) and *qacA*-positive isolates (Type V, 4/13) in the C2A subclade (Supplementary Data 1). Moreover, *qacA*-negative isolates formed two large C4 and C6 (Australian) subclades and a small C5 (American) subclade. The isolates in these *qacA*-negative subclades belonged to SCC*mec* type V (100%, 152/152) and lacked *aac(6')-aph(2")* (0%, 0/152), but were frequently associated with *tetK* (83.4%, 126/152) and less often carried *ermC* (49.3%, 75/152). The available metadata on the hospital and community origin of these isolates was limited (Supplementary Data 1). *qacA* was detected in 45.7% (16/35) of the community-associated isolates and 52% (13/25) of the hospital-associated isolates. Although 98.4% (379/385) of the isolates were MRSA,

**Fig. 1 | Global phylogeny of ST45 *S. aureus* and *qacA* carriage. a** Maximum-likelihood tree was constructed using 37,549 core SNPs identified in the global collection of 2001 ST45 *S. aureus*. **b** Maximum-likelihood tree of 616 ST45 *S. aureus* in BAPS 4 was built using an alignment of 4716 core genome SNPs. From the innermost to the outermost ring, the heatmaps display (i) population clusters or sub-clusters determined by hierBAPS, (ii) whether an isolate was MRSA or MSSA based on *mecA* carriage, (iii) presence or absence of *qacA* carriage, (iv) geographical origin of the isolates. Isolates with unknown geographic information were denoted by the blank heatmaps on the outer ring.

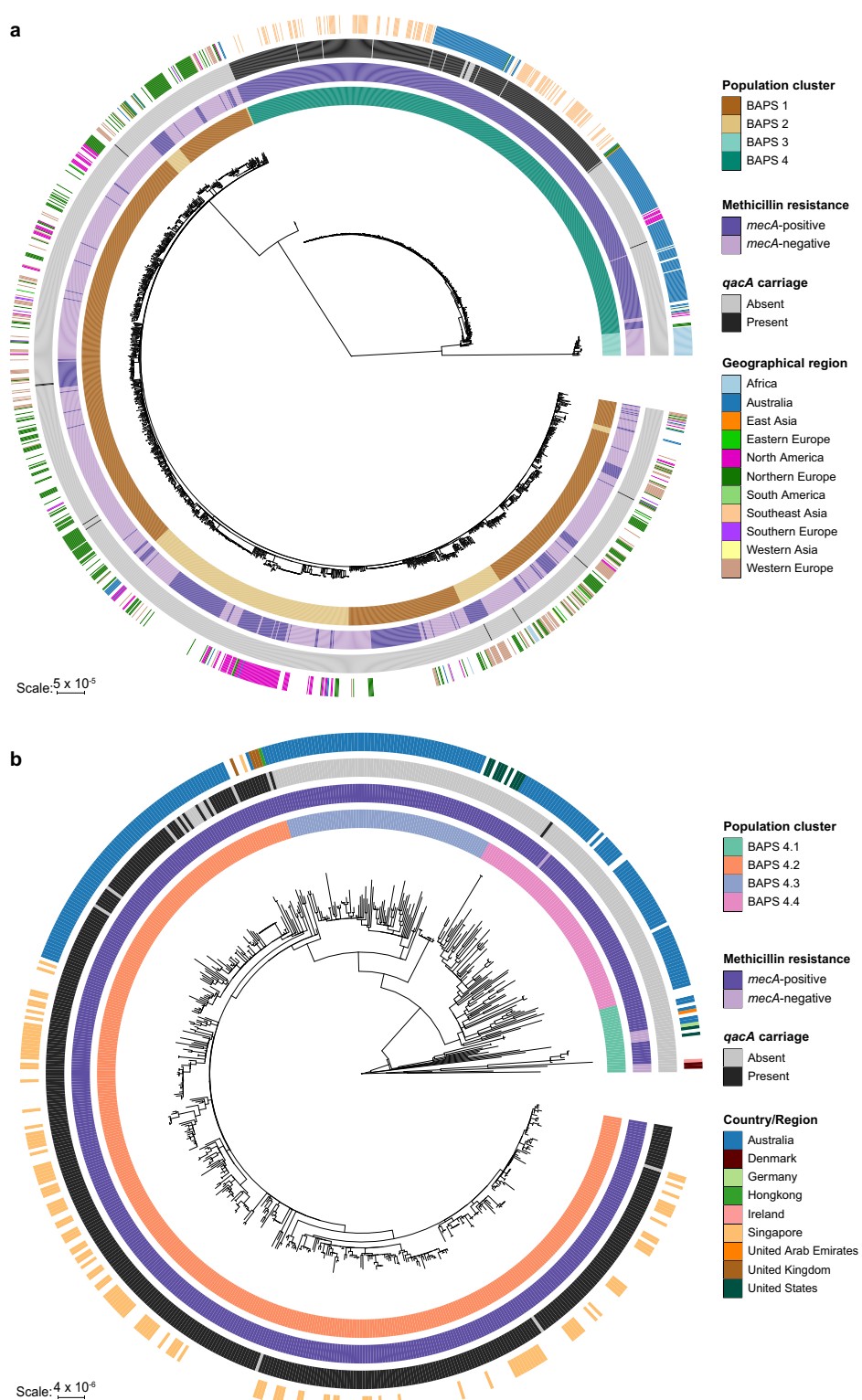

the most recent common ancestor for the BAPS 4 clade likely arose in the 1960s, with the oldest isolate found in the C7 (European) subclade being a methicillin-susceptible *S. aureus* (MSSA) of Danish origin, collected in 1970 (Fig. 2a).

To determine the temporal change in the population structure and size of ST45 MRSA subclades in relation to *qacA* carriage in Australia, the effective population size (EPS) and effective reproduction number ($R_e$) of major *qacA*-positive (C2B) and *qacA*-negative (i.e., C4 and C6) subclades were compared. Of note was the rapid increase in the median EPS of C2B between 2008 and 2018 (Fig. 2b), with the first peak observed in 2014

attributed to the Singaporean isolates. Extended analysis using the entire C2 subclade showed a population collapse in 2019 due to the inclusion of C2A subclade (Supplementary Fig. 1). In comparison, the median EPS of C4 appeared to have increased between 2007 and 2014, followed by a reduction from 2015 onwards. However, the broad 95% HPD interval indicates large uncertainty in estimating the EPS trajectory of this subclade prior to 2015. Similarly, the median EPS of the C6 subclade steadily increased until 2012, when it started to plateau and then declined between 2014 and 2018. The 95% HPD intervals of EPS of all three subclades overlapped between 2015 and 2018. Consistently, epidemic spikes in $R_e$ (95% HPD > 1, indicating

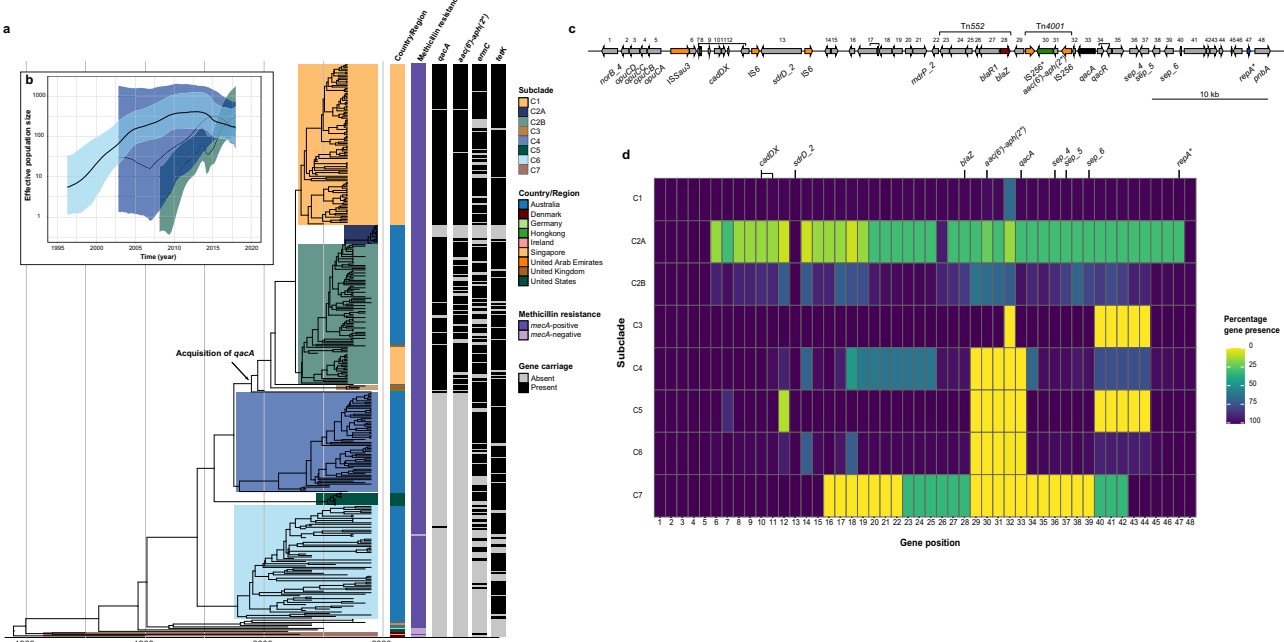

**Fig. 2 | Evolutionary history of *qacA*-harbouring ST45 MRSA. a** Maximum clade credibility tree of 385 ST45 *S. aureus* isolates. The heatmaps provide additional annotations regarding country/region of isolation, methicillin resistance determined by *mecA* carriage, and the presence or absence of additional AMR determinants, including *qacA*, *aac(6')-aph(2")*, *emrC*, and *tetK*. The black arrow indicates the predicted time for *qacA* acquisition (median node age = 1998, 95% HPD = 1996 to 1999). These isolates were divided into seven subclades based on *qacA* carriage and geographic origin. **b** GMRF Bayesian skyride plot. EPS of the major *qacA*-positive C2B (light green) subclade was compared with those of the *qacA*-negative C4 (navy blue) and C6 (light blue) subclades. The solid black line indicates the median EPS, and the coloured boundary indicates the 95% HPD. **c** The chromosomal region

surrounding *qacA* in AUSMDU00020487. Each CDS (grey arrow) in this region was assigned a positional number, and CDSs of the same ortholog groups are linked by black solid lines. ISs (orange), plasmid replication initiator gene *repA* (blue), and AMR genes, including *blaZ* (red), *aac(6')-aph(2")* (green), and *qacA* (black), are highlighted. Boundaries of Tn*552* and TN*4001* are depicted by the brackets. Fragmented genes are flagged with an asterisk. **d** Comparison of gene content surrounding *qacA* between the major subclades. Percentages of gene presence are displayed in the heatmap, with dark blue indicating the presence of the gene among all isolates within a given subclade, while yellow indicates the absence of the gene. A summary of gene functionality, dosage, and synteny can be accessed in Supplementary Data 3.

exponential spread) were observed during the emergence of C2B (2011–2013) and C4 (2010–2012) as well as in recent years for C4 (2014–2017) and C6 (2013–2017) (Supplementary Fig. 2). In contrast, $R_e$ decreased or stabilised in C2B (95% HPD overlapping $R_e = 1$, between 2013 and 2017). Collectively, these results demonstrate an overall increase in the population size of ST45 MRSA, regardless of *qacA* status over the past two decades, but dynamic changes in population growth were noted, including periods of likely exponential spread. However, there was no clear association between *qacA* acquisition and epidemic growth of *qacA*-harbouring isolates in the C2B subclade.

### Chromosomal integration of *qacA* in ST45 MRSA

The presence of a chromosomal *qacA* (rather than plasmid-associated) in ST45 MRSA prompted further investigation into the genetic context of *qacA*. Gene conservation (Fig. 2c) and synteny (Supplementary Fig. 3) of each subclade were compared. The genomic configuration in all isolates belonging to subclades C4 – C7 showed early acquisition of multiple genetic elements, including *mdrP* (encoding a Na + /H+ antiporter), Tn*552* carrying *blaZ*, and three enterotoxin encoding genes (*sep*) (Fig. 2c). These elements were likely brought into this region of the chromosome by the integration of a pF5-like plasmid, suggested by: (i) a fragmented *repA* gene, and (ii) a high level of similarity (>97%) to the genomic structure of a previously reported pF5 plasmid (GenBank accession number: AB765928) lacking Tn*552* (Supplementary Fig. 4). The plasmid reported in *S. aureus* strain Fukuoka 5 has previously been associated with food poisoning-related illnesses[29]. Further, it was evident that changes to this region have occurred since the possible integration of a pF5-like plasmid, with the loss of a hypothetical protein CDS cluster upstream of *repA* in the C3 and C5 subclades (Fig. 2d, gene position 40–44).

Acquisition of *qacA*, possibly concurrent with Tn*4001* containing *aac(6')-aph(2")*, occurred in a MRSA background carrying *blaZ* as displayed in the C1, C2, and C3 subclades (Fig. 2d, gene position 29–33). Sequence alignment between AUSMDU00020487 and BPH2770 (GenBank accession number: GCA_027956625.1), a *qacA*-negative ST45 MRSA reference, revealed a 7981 bp cassette containing *qacA* and Tn*4001* (Supplementary Fig. 5). This cassette shared 99% homology with the Tn*4001-qacR* spanning region of a pSK1 plasmid (GenBank accession number: NC_014369) with an inverted *qacAR* (Supplementary Fig. 4). However, the pSK1 plasmid did not contain the small putative membrane protein-encoding gene found between *qacA* and Tn*4001*, nor the hypothetical protein coding sequence between the truncated IS*256* and *blaZ* (Supplementary Fig. 4). We did not identify evidence of other ISs or transposable elements flanking the putative cassette, and the mechanism for chromosomal integration remains unclear. Of note, this region was highly variable within the C2 subclade, in which the genes were less syntenic (Supplementary Fig. 3). Some isolates in the C2A subclade had lost the region (51,010 bp) spanning between IS*Sau3* and the fragmented *repA*, suggesting a potential IS*Sau3* transposition (Supplementary Fig. 6). These findings highlight the instability of the chromosomal region surrounding *qacA*, as the gain and loss of genes or mobile genetic elements appeared to have frequently occurred.

### *qacA* in ST45 MRSA confers tolerance to quaternary ammonium compounds

To investigate the potential role of *qacA* in conferring tolerance to QACs in ST45 MRSA, a markerless deletion of *qacA* was made in the reference strain, AUSMDU00020487, using a previously described method[30]. Broth microdilution MIC assays showed that the deletion of *qacA* led to a twofold reduction in the chlorhexidine digluconate (CHG) MIC (from 2 to 1 mg/L)

and a 16-fold reduction in acriflavine MIC (from 128 to 8 mg/L) compared to the wild-type strain (Table 1). The ratio of the minimum bactericidal concentration (MBC) to MIC was 2, as CHG and acriflavine MBCs were 2 mg/L and 16 mg/L for the ΔqacA strain, respectively; While CHG and acriflavine MBCs were 4 mg/L and 256 mg/L for the wild-type strain, respectively. Complementation of qacA restored the MICs and MBCs to the levels of the wild-type (Table 1). No secondary mutations confounding the observed phenotypic changes were detected in the ΔqacA and ΔqacA:qacA strains (Supplementary Data 2). The MIC and MBC results were consistent across three biological replicates.

### In vitro exposure to sub-MIC of chlorhexidine selects for qacA-harbouring ST45 MRSA

Competition assays were performed to determine the impact of qacA in conferring a competitive advantage in the presence of CHG, as previously described[31]. Co-cultures of the AUSMDU00020487 ΔqacA strain paired with either the wild-type or ΔqacA:qacA strain were established in a 1:1 ratio. These co-cultures were supplemented with or without CHG at a sub-MIC level (0.5 mg/L) for the ΔqacA strain. The results demonstrated that qacA-harbouring strains rapidly outcompeted the ΔqacA strain, with 100% of the isolates recovered from the co-cultures being wild-type or ΔqacA:-qacA following 24 h exposure to CHG (day 1) (Fig. 3). The qacA-harbouring strains remained dominant at the conclusion of the assays on day 7, with 100% and >99% of the harvested isolates being wild-type and ΔqacA:qacA strains, respectively (Fig. 3). In contrast, no noticeable variation in the proportion of wild-type or ΔqacA:qacA strains relative to the mutant was observed on day 1 or day 7 in the absence of selective pressure. Additionally, the doubling time of the ΔqacA strain, ΔqacA:qacA and wild-type strains did not significantly differ ($P > 0.05$ determined by unpaired $t$ test,

Supplementary Fig. 7), indicating that the sub-MIC selection of qacA-harbouring strains was not associated with a faster bacterial growth rate.

## Discussion

In this study, we investigated an emerging qacA-harbouring ST45 MRSA lineage using genomic and phenotypic analyses. In agreement with a recent study examining the emergence of ST45 *S. aureus*[22], our analysis using a larger dataset revealed a clonally diverse global population structure with SCC*mec* Type V MRSA predominated among Australian isolates. Within the global dataset, we show that the presence of qacA was almost exclusively found in ST45 isolates from Australia and Singapore, with ST45 becoming a predominant MRSA lineage in these regions over the past decade[16,32]. Noticeably, a high prevalence (99%) of qacA has previously been reported in ST45 MRSA linked to universal chlorhexidine bathing implemented in Singaporean extended-care facilities[12]. However, the prevalence of qacA and its potential role in the evolutionary success of ST45 MRSA has not previously been investigated in Australia.

We observed a high carriage rate of qacA among ST45 MRSA isolates collected from the national staphylococcal bacteriemia surveillance programmes undertaken in 2015[26] and 2017[27]. qacA-harbouring isolates in our study shared a similar AMR profile (*aac(6′)-aph(2″)*, *ermC*, and *tetK*) with those responsible for local hospital outbreaks in the state of NSW between 2013 and 2017[19]. Outside Australia, ST45 MRSA has been associated with healthcare settings in Singapore[12] and Taiwan[33], suggesting possible intercontinental dissemination of this lineage.

Phylodynamic analysis suggested that ST45 MRSA expanded over the past two decades, including qacA-harbouring sublineages. Population expansion was also observed in older lineages harbouring *ermC* and *tetK*, but lacking qacA. It is possible that the early introduction of these AMR determinants (i.e., *ermC* and *tetK*) into ST45 MRSA might have conferred an initial selective advantage for this lineage, with the subsequent acquisition of qacA and *aac(6′)-aph(2″)* in the late 1990s providing a further basis for selective expansion after 2013[18]. The increased prevalence of ST45 MRSA coincided with the increase in chlorhexidine-based infection control and hand hygiene programmes designed predominantly to combat MRSA[34]. Dynamic changes in the $R_e$ occurred in both qacA-positive and qacA-negative sublineages, indicating that the sizes of these populations were fluctuating. While the acquisition of qacA may be an important contributor to the latest regional outbreaks caused by ST45 MRSA, further investigation using contemporary isolates is needed to evaluate the ongoing impact of

**Table 1 | In vitro susceptibility to quaternary ammonium compounds in AUSMDU00020487 and the derived mutants**

| Strain | Genotype | CHG (mg/L) | | Acriflavine (mg/L) | |
|---|---|---|---|---|---|
| | | MIC | MBC | MIC | MBC |
| AUSMDU00020487_wild-type | Wild-type | 2 | 4 | 128 | 256 |
| AUSMDU00020487_qacA | ΔqacA | 1 | 2 | 8 | 16 |
| AUSMDU00020487_qacA:qacA | ΔqacA:qacA | 2 | 4 | 128 | 256 |

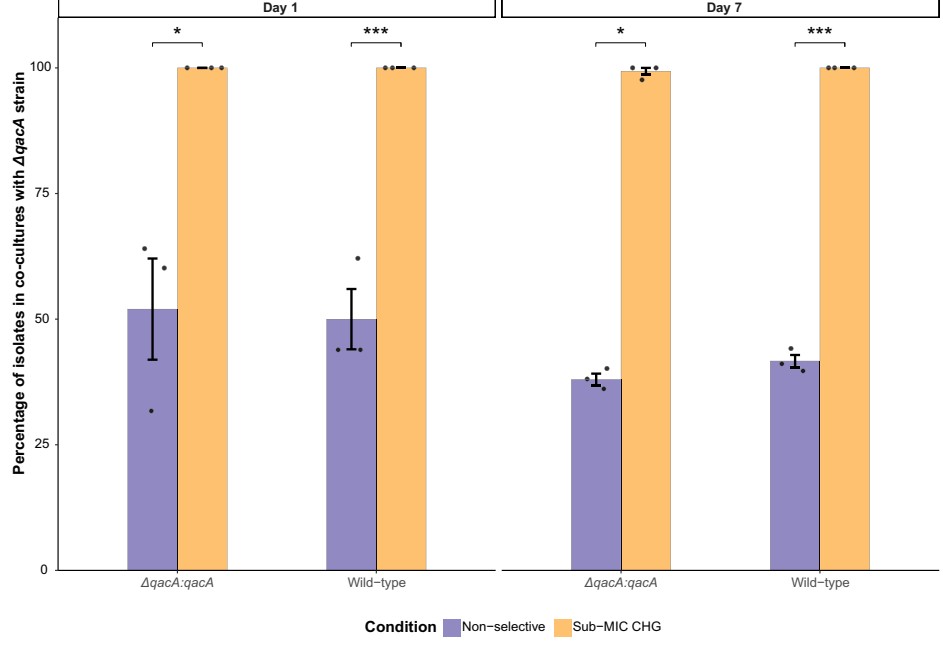

**Fig. 3 | In vitro competition assays using co-cultures of AUSMDU00020487 wild-type paired with ΔqacA strain, and ΔqacA:qacA paired with ΔqacA strain.** These co-cultures were exposed to either non-selective (purple) or selective condition using 0.5 mg/L CHG (yellow) for 7 days. Three biological replicates denoted by the black dot points were performed for each condition tested. The mean percentages of wild-type or ΔqacA:qacA strains on day 1 and day 7 post-exposure are displayed, with the black error bars representing the standard error of the mean. Asterisks denote statistically significant differences as determined by paired $t$ test (*$P \leq 0.05$ and ***$P \leq 0.001$).

these sublineages (both *qacA*-positive and *qacA*-negative) in contributing to the burden of *S. aureus* infections in Australia.

*qacA* was co-located with multiple AMR and virulence determinants in the chromosome of ST45 MRSA, similar to the chromosomal *qacA* integration in ST239 MRSA[15]. Comparative genomic analysis showed that early integration of a pF5-like plasmid (characterised by enterotoxin encoding genes) in historic ST45 isolates may have provided a 'hotbed' for subsequent acquisition of *qacA* and other mobile elements, including Tn*552* and Tn*4001*. The gain of these mobile elements may also confer additional evolutionary benefits by co-selection of AMR (namely, *blaZ* and *aac(6')-aph(2")*). However, the inference drawn from this analysis was limited by the lack of isolates collected before 2009 (the year in which the first *qacA*-harbouring isolate was reported in our dataset), which may have impacted the accuracy of our prediction regarding the temporal emergence of *qacA*. Among the recent isolates collected in 2019, we observed a loss of regions surrounding *qacA*, possibly driven by IS*Sau3*. This may provide opportunities for further spread of *qacA* and other co-located AMR and virulence determinants. Given chlorhexidine use has been linked to the enrichment of *qacA* carriage[35–37], IS-mediated transposition could be reflective of demographic differences in infection control practices across geographic settings.

Our mutagenesis work provided experimental evidence that the presence of *qacA* was associated with increased tolerance to CHG, when the ST45 reference strain (MIC = 2 mg/L) was compared to its Δ*qacA* strain (MIC = 1 mg/L). However, the shift in MIC is below the cut-off (MIC ≥ 4 mg/L) used for determining chlorhexidine tolerance in some studies[38,39]. This underscores a known issue with using strict MIC cut-offs, in that subtle shifts in tolerance may be missed. The use of a higher cut-off might have led to the observation that *qacA* carriage in ST45 MRSA is not always associated with reduced susceptibility to chlorhexidine[12]. This inconsistency might be caused by the assay itself, which was originally intended for antibiotic susceptibility testing, and caution should be advised when used for biocide testing[40]. Of note, we observed that acriflavine may be a useful alternative for differentiating *qacA*-positive (MIC = 128 mg/L) from *qacA*-negative (MIC = 8 mg/L) isolates in vitro. Together, our findings and previous observations collectively emphasise the need for an improved and standardised methodology (validated across multiple laboratories) for determining biocide tolerance.

The demonstration of sub-MIC selection provides evidence that *qacA* may be an important determinant in enhancing the adaptation of ST45 MRSA to low levels of CHG exposure. While the in-use concentrations of chlorhexidine (e.g., 0.5, 2, and 4% (w/v)) are considerably higher than the sub-MIC tested[41], it is worth noting that the selection of *qacA*-harbouring isolates has been reported in ST5, ST22, and ST239 MRSA backgrounds following chlorhexidine-based infection control measures[42–44]. Of concern is the co-selection of AMR and virulence genes as an unintentional consequence of chlorhexidine use. As such, this finding has implications for understanding the prolonged residual effect of chlorhexidine on enriching for *qacA* carriage as well as increasing the overall drug resistance in *S. aureus*.

This study has several strengths. First, the sample size for our global phylogenetic analysis was large and included representative isolates from various geographic regions. Second, the complete genome of a local reference strain was constructed to provide detailed insights into the emergence and divergence of major sublineages in Australia. Third, mutagenesis and in vitro assays were used to characterise *qacA*-mediated biocide tolerance in the reference strain.

There are limitations to our sampling strategy. First, the missing geographic information for a large number of isolates included in the global phylogeny of ST45 *S. aureus* and the overrepresentation of Australian and Singaporean isolates sampled across limited time frames should be noted, limiting our capacity to fully understand the global emergence and dissemination of this lineage. Second, due to the lack of historic isolates, we were unable to fully elaborate on the sequential order of integration events leading to the chromosomal integration of *qacA*. Third, a population-level analysis of biocide tolerance, which could provide critical insights into the phenotypic contribution of *qacA* to biocide tolerance in different ST45

MRSA sublineages, was not assessed in this study. Fourth, we did not possess metadata regarding the community or hospital origin for the majority of isolates included in the phylogenetic analyses (Supplementary Data 1). This limitation limited our ability to estimate the burden of *qacA*-harbouring isolates and understand their adaption to settings with routine antiseptic use. Thus, future work should focus on (i) understanding the causative link between contemporary biocide usage and population-level phenotypic tolerance; (ii) evaluating the stability of the *qacA*-containing region in the chromosome as previously described[45]; (iii) determining the impact of clinically used concentrations of chlorhexidine in selecting for *qacA* carriage; and (iv) enhancing Bayesian phylogenetics on investigating origin and dissemination of the *qacA*-harbouring lineage/sublineages using additional historic and contemporary isolates with diverse geographic background. Regionally, the integration of epidemiological data and improved knowledge of regional differences in antiseptic use will also facilitate our understanding of whether AMR and chlorhexidine tolerance are key drivers in the selection of ST45 MRSA in Australia.

Despite decades of biocide use, adverse ecological impact associated with the widespread use of chlorhexidine has raised questions about the need to develop biocide stewardship. Here, we have described the emergence and clonal expansion of a *qacA*-harbouring ST45 MRSA lineage. Together with the phenotypic confirmation of its role in mediating chlorhexidine tolerance, our data suggest that *qacA* may have important implications for the recent clonal spread of ST45 MRSA in Australia and Asia.

## Methods

### Bacterial isolates and sequence data
*S. aureus* genomes used in this study were sourced from Staphopia[28], European Nucleotide Archive (ENA) per the study conducted by Effelsberg et al.[22], and surveillance programmes and national studies of *S. aureus* undertaken in Australia[23–27]. Further demographic information regarding the sources of isolates and selection criteria for sequence data are summarised in the Supplementary Information (Supplementary Table 1).

### Construction of reference genome
A *qacA*-positive ST45 MRSA strain, AUSMDU00020487, was collected in Victoria, Australia, in 2018[23]. To generate a complete reference genome, genomic DNA was sequenced on the Illumina NextSeq or iSeq (2 × 150 bp paired-end chemistry) and Oxford Nanopore GridION X5 (with FLO-MIN106D R9 flow cells) platforms. Demultiplexing and adaptor trimming of the Nanopore long-read data was performed using Porechop v0.2.4 (https://github.com/rrwick/Porechop). Trimmed data were filtered using Filtlong v0.2.1 (https://github.com/rrwick/Filtlong) by removing short reads (<1000 bases), and sampling to a target total bases value of 300 megabases, favouring the longest and highest quality reads. A hybrid assembly was then undertaken with Unicycler v0.4.8[46]. The assembled genome underwent further error correction with the Illumina short-read data using Snippy v4.4 (https://github.com/tseemann/snippy) iteratively until no variants were called. The draft genome was annotated with Prokka v1.14.6[47] and analysed using mlst v2.19.0 (https://github.com/tseemann/mlst) to confirm that AUSMDU00020487 belonged to ST45 *S. aureus*. AMR genotyping analysis was performed using ABRicate v0.8.10 (https://github.com/tseemann/abricate) against the National Center for Biotechnology Information (NCBI) AMRFinderPlus database[48]. SCC*mec* typing was performed using staphopia-sccmec v1.0.0[28], and SCCmecFinder v.1.2[49] was used to distinguish between SCC*mec* Type V and VII.

### Comparative genomic analysis
Illumina short reads or de novo assemblies of the global collection of ST45 *S. aureus* were mapped to the reference AUSMDU00020487 genome, using Snippy v4.4. Core SNPs identified were then used for constructing the maximum-likelihood phylogenetic tree with IQ-TREE v2.1.2[50], under a generalised time-reversible (GTR) model combined with the proportion of invariable sites, empirical base frequencies, a discrete gamma model with 4 rate categories (GTR + I + F + G4) and 1000 bootstrap replicates. BAPS

clusters were identified using hierBAPS v1.0.1[51] with two levels and ten initial clusters on the core genome SNP alignment. All trees were midpoint rooted and visualised using R package *ggtree* v3.6.2[52].

Short reads were de novo assembled using SPAdes v3.1.2[53] and annotated by Prokka v1.14.6. MLST typing was performed on the genome assemblies with mlst v2.19.0. ABRicate v0.8.10 with the NCBI's AMRFinderPlus database was then used for determining the AMR profile for each isolate. Isolates were categorised into MRSA or MSSA based on *mecA* carriage. Automated clustering of protein orthologs was performed using Roary v3.13.0[54] with a 95% amino acid homology. Gene content, synteny, and dosage were compared between subclades. Annotation for insertion sequences was checked by comparing the predicted IS CDS against the ISfinder database[55].

## Phylogenetic analysis

A core genome SNP alignment of 385 ST45 *S. aureus* was built, as described above. Recombination detection was performed using ClonalFrameML v1.12 (https://github.com/xavierdidelot/ClonalFrameML). Recombination sites, including 22 core SNPs, were masked using maskrc-svg v0.5 (https://github.com/kwongj/maskrc-svg). The strength of the temporal signal in the masked alignment was assessed using TempEst v1.5.3[56]. To determine the optimal parameters for Bayesian phylogenetics, the masked alignment containing 3690 filtered core SNPs underwent further analysis using BEAST v1.10.4[57] under a GTR + G4 model of nucleotide substitution, with combinations of the strict and uncorrelated relaxed log-normal molecular clocks, and constant and exponential tree priors. The XML files generated by BEAUti were edited to reflect the number of invariant sites in the masked alignment. For each combination tested, marginal likelihood estimation was performed in triplicate using path sampling to calculate the Bayes factors. The best-fitting model was achieved by the uncorrelated relaxed log-normal clock with a coalescent exponential population tree prior. Using this parameter, ten replicates of 50 million Monte Carlo Markov Chain (MCMC) analyses were performed with sampling every 1000 steps and 20% burn-in removed. Replicates were combined using LogCombiner and re-sampled at a frequency that would achieve ~10,000 samples. Tracer v1.7.2[58] was used to check MCMC convergence and effective sample size value (greater than 200) to ensure adequate sampling of the posterior distribution. The maximum clade credibility tree with median heights was selected with TreeAnnotator.

## Population genetic analysis

Subsets of the main core SNP alignment were created for several subclades of the above maximum clade credibility tree. These subclades were selected due to the number of available isolates, serial sampling dates, and their relevance to the emergence of ST45 in Australia. Following the same steps above for Bayesian phylogenetics, EPS was estimated using the uncorrelated relaxed log-normal clock coupled with a time-aware Gaussian Markov random field (GMRF) Bayesian skyride model[59]. The Bayesian skyride plots of median EPS were generated in Tracer v1.7.2[58] using the GMRF skyride reconstruction tool. Furthermore, serial birth–death skyline models in BEAST v2.6[60,61] were configured to infer changes in transmission dynamics at sublineage resolution. For all subclades, the model was run under a GTR + G4 model of nucleotide substitution using a strict clock model for simplicity, which was chosen due to the short timeframe of emergence (8–14 years) precluding assumptions of significant rate variation amongst clade members. Model priors, including a sliced effective reproductive number prior, were configured as described previously for *S. aureus*[62]. In brief, the $R_e$ prior was configured to a number of equally sized intervals over the maximum clade credibility tree; a suitable interval number was selected by running exploratory models for each lineage with 100 million iterations (dimensions = 5–10). A comparison of $R_e$ estimates under these configurations ensured the occurrence of stable posterior distributions and the absence of bi- or multimodal posteriors. Five equally sliced dimensions were then selected as a parsimonious configuration for the prior, allowing each slice to cover a range of 2–3 years depending on clade sampling times. Five

replicates of 100 million MCMC steps with slight variation in the initial parameter values were run with sampling every 1000 steps and 20% burn-in removed. Tracer was used to check MCMC convergence and to ensure adequate sampling of the posterior distribution. Critter (https://github.com/esteinig/critter) was used to configure the models and to generate the plots depicting changes of average $R_e$ across slices and kernel density esimates of the underlying posterior distribution across slices (Supplementary Fig. 2).

## Construction of Δ*qacA* strain by allelic exchange

To make a markerless deletion of *qacA* in AUSMDU00020487, a 1.5-kb gene cassette containing jointed flanking regions upstream and downstream of *qacA* was amplified by spliced overlap extension PCR. The cassette was cloned into the pIMAY-Z shuttle vector[30] using seamless ligation cloning extract[63]. The cloned plasmid was electroporated into *Escherichia coli* IM30B to obtain methylation profiles compatible with ST45 *S. aureus*. Successful *E. coli* transformants were selected following overnight incubation at 37 °C on Luria agar (LA) containing 10 mg/L of chloramphenicol and 50 mg/L of X-gal (5-bromo-4-chloro-3-indo-lyl-D-galactopyranoside; Melford). Extracted plasmid from *E. coli* transformants was then introduced into AUSMDU00020487 by electroporation. AUSMDU00020487 transformants (indicated by blue colonies) were passaged in brain heart infusion (BHI) broth at 30 °C (i.e., non-permissive temperature for plasmid replication) to promote chromosomal integration of pIMAY-Z. To perform allelic exchange, dilutions of the broth culture were spread onto BHI agar supplemented with 100 mg/L of X-gal, followed by overnight incubation at 37 °C. White colonies were cross-patched onto BHI agar containing 10 mg/L of chloramphenicol and 100 mg/L of X-gal; and BHI agar containing 100 mg/L of X-gal only. Following overnight incubation at 37 °C, white colonies displaying susceptibility to chloramphenicol (indicating loss of pIMAY-Z plasmid) were confirmed as mutants by PCR and whole genome sequencing. For *qacA* complementation, a substitution at nucleotide 720 (c.720 A > G) resulting in a silent mutation at codon 240 (p. Pro240Pro) was introduced to the Δ*qacA*:*qacA*. Secondary mutations in the Δ*qacA* and Δ*qacA*:*qacA* strains were screened using Snippy v4.4 (Supplementary Data 2). All primers used in the experiments are listed in Table 2.

## Antimicrobial susceptibility testing and growth assay

Broth microdilution MIC and MBC assays were performed in accordance with the Clinical & Laboratory Standards Institute (CLSI) guidelines[64] using 20% CHG solution (C9394; Sigma-Aldrich) or acriflavine (A8126; Sigma-Aldrich). Briefly, a twofold serial dilution of an antimicrobial was performed in 100 μL of Cation-adjusted Mueller Hinton (CAMH) broth in a 96-well microplate. A fresh bacterial culture was used to prepare a 0.5 MacFarland suspension in CAMH broth, and 100 μL of the suspension was dispensed into the wells containing the antimicrobial. The microplate was incubated at 37 °C overnight before MIC determination. MIC was defined as the lowest antimicrobial concentration inhibiting observable bacterial growth. To perform MBC testing without neutralisation of the antimicrobial, the contents in each well of the microplate was mixed thoroughly. 10 μL of the resuspension was spot-plated onto CAMH agar, which was incubated for 48 h at 37 °C. MBC was defined as the lowest antimicrobial concentration required to prevent bacterial growth on agar. Three biological replicates were performed for both MIC and MBC testing.

Growth assays were performed as previously described[65]. Briefly, an overnight bacterial broth culture was diluted in fresh BHI broth to an optical density of 0.05 at 600 nm (OD$_{600}$). Subsequently, 200 μL of the bacterial suspension was distributed into a 96-well microplate. The microplate was incubated in a microplate reader (CLARIOstar, BMG LABTECH) at 37 °C for 16 h with agitation at 200 rpm, and the OD$_{600}$ was measured at 15-min intervals. Six biological replicates were performed for each strain tested. The bacterial growth rates denoted as doubling times were determined using the R package *Growthcurver* v0.3.1[66].

**Table 2 | Primers used in this study**

| Primer | Sequence (5' – > 3') | Description |
|---|---|---|
| qacA_Fp | CCTCACTAAAGGGAACAAAAGCTGGGTACCCTTTTAATTCTAGCGTGCCTAC | Flanking primers used for amplification of gene cassettes for *qacA* deletion and complementation |
| qacA_Rp | CGACTCACTATAGGGCGAATTGGAGCTCCGTAATTTAGAAATAATATTTATTGGTATTTCAAG | |
| qacA_SOE_Fp | CTAATCTACAATATCTAAAAATATATGTTTAGTATTAAGTTCCCTCCAATCCTTATAG | Construction of gene cassette for *qacA* deletion by Splicing by Overlap Extension PCR |
| qacA_SOE_Rp | CTATAAGGATTGGAGGGAACTTAATACTAAACATATATTTTTAGATATTGTAGATTAG | |
| qacA_comp_Fp | CTATTACAACCCACGGAATAATATCTGCTAGTC | Construction of gene cassette for *qacA* complementation by Splicing by Overlap Extension PCR |
| qacA_comp_Rp | GACTAGCAGATATTATTCCGTGGGTTGTAATAG | |

## In vitro competition assay

To determine the competitive advantage conferred by *qacA* carriage following exposure to CHG, an in vitro assay was performed according to a previously established protocol[31]. This assay tested the pairing of (i) AUSMDU00020487 wild-type and Δ*qacA* strain, and (ii) Δ*qacA*:*qacA* and Δ*qacA* strain. These strains were grown in BHI broth overnight prior to the assay. In brief, an overnight bacterial culture was first diluted with fresh BHI broth to an $OD_{600}$ of 0.10. A bacterial suspension was prepared by mixing the competing strains in a 1:1 ratio. The mixed suspension was then diluted 1:100 in 10 mL (i) non-selective BHI broth or (ii) BHI broth supplemented with 0.5 mg/L of CHG (i.e., $0.5 \times MIC$ for the Δ*qacA* strain). The co-cultures were incubated at 37 °C with shaking at 200 rpm for 7 days. Following 24 h of incubation (day 1), a 300 µl sample was withdrawn and tenfold serially diluted in phosphate-buffered saline. In total, 100 µl of appropriate dilutions were spread onto BHI agar and incubated at 37 °C overnight. Following the incubation, 50 single colonies were randomly selected and cross-patched onto cation-adjusted Mueller Hinton agar supplemented with or without 64 mg/L of acriflavine to determine the ratio of the two competing strains on day 1. The same process was repeated on day 7 post-exposure. Three biological replicates were conducted for each pair/condition tested. For strain validation and analysis of secondary mutations, two output isolates collected at the conclusion of the experiment under the selective condition were selected for Illumina short-read sequencing (Supplementary Data 2). The results were visualised using R package *ggplot2* v3.4.1[67].

## Statistics and reproducibility

Unpaired and paired two-sided *t* tests (two-tailed) were performed using R package *rstatix*[68] to compare bacterial growth rate and in vitro competition, respectively. A *P* value lower than 0.05 was determined to be statistically significant.

## Reporting summary

Further information on research design is available in the Nature Portfolio Reporting Summary linked to this article.

## Data availability

Sequence data for the isolates retrieved from publications and public databases can be accessed in accordance to Supplementary Note 1 and Supplementary Data 1. Additional genomic data generated in this study have been deposited under the BioProject PRJNA984755 in the NCBI database. The complete genome assembly of the ST45 *S. aureus* reference AUSMDU00020487 strain (GenBank accession numbers: CP138566-7) has been uploaded under the BioProject PRJNA565795. Numerical source data for competition assays, growth assays, and $R_e$ analysis are available on Figshare (https://doi.org/10.26188/25201073).

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

## Acknowledgements

The authors would like to thank the staff in the Microbiological Diagnostic Unit Public Health Laboratory, Victoria, Australia and in the PathWest Laboratory Medicine, Western Australia, Australia. This work is funded by a National Health and Medical Research Council (NHMRC) Investigator Grant (APP1174555).

## Author contributions

D.A.W., B.P.H., G.P.C., S. Pasricha, S.L.B., and Y.N. contributed to the conception and design of the work. G.W.C., D.A.D., and S. Pang provided the genomic data from the Australian Staphylococcal Sepsis Outcome Program (ASSOP). P.N.A.H. and B.M.F. provided the genomic data from the Queensland surveillance of MRSA program. B.P.H. and S.L.B. provided genomic data from the Australian and New Zealand Cooperative on Outcome in Staphylococcal Sepsis (ANZCOSS) study and the Vancomycin Efficacy in Staphylococcal Sepsis in Australasia (VANESSA) study. B.P.H. and N.L.S. provided the genomic data and reference isolates from the 'Controlling Superbugs' study. G.P.C., I.R.M. and Y.N. designed the experimental procedures and provided materials. Y.N. performed the genomic and phylogenetic analyses with input from E.S. and guidance from S.L.B. Y.N. performed experiments with input from G.L.P. and guidance from G.P.C. and I.R.M. D.A.W., G.P.C., S. Pasricha, S.L.B., G.L.P., E.S., G.T. and Y.N. contributed to the interpretation of the results. Y.N. conducted the statistical analyses, produced the figures and tables, and drafted the manuscript. All authors revised and approved the manuscript.

## Competing interests

The authors declare no competing interests.
