## [Peer Review File · Communications Biology]

Reviewers' comments:

Reviewer #1 (Remarks to the Author):

The current study presents a bioinformatics analysis of a large collection of *S. aureus* ST45 and reports the presence of *qac* genes, which is known to confer resistance to biocides. Moreover, the authors present a timed phylogeny and are able to predict, with confidence, when the acquisition of the *qac* genes happened in ST45. The bioinformatics analyses are supplemented by *in vitro* experiments, in which authors demonstrate that indeed the presence of *qac* genes is responsible for decreased susceptibility to chlorhexidine, a biocide widely used in clinical settings.

The bioinformatics methods employed in the current study are sound and state of the art, however, the information presented in the manuscript is not novel, in the sense that presence of *qac* genes in MRSA and ST45 in particular has been widely reported before.

For example, an Irish study on samples collected between 2000-2012

<https://doi.org/10.1128/aac.02653-13> has reported *qac* genes in various *S. aureus* sequence types.

Another paper from 2011 (<https://doi.org/10.1128/aac.02653-13>) on the prevalence of *qac* genes in samples from nurses and general population in HK found that 50% of MRSA isolates carry *qac* genes.

A 3-year study on 878 isolates from Singapore (<https://doi.org/10.1016/j.cmi.2018.12.036>, not cited in the manuscript currently under review) found that

"Among *qacA/B* carrying MRSA, the majority of *qacA/B* was found in ST45 (n = 287/409; 70.2%), followed by ST22 (n = 71/409; 17.3%), and other MRSA STs (n = 51/409; 12.5%)."

It is my belief that the timed phylogeny and the results of the *in vitro* experiments presented in this manuscript do not add enough novel information for the manuscript to be published in the current journal.

Reviewer #2 (Remarks to the Author):

The article by Nong and colleagues explores the dissemination of ST45 *Staphylococcus aureus* from a phylogenomic and phenotypic perspective, and attempt to understand the role of the acquisition of *qacA*, a resistance determinant involved in tolerance to biocides. The study utilizes a well-structured design and employs various techniques, including phylogenomic and population dynamic analyses, to investigate a robust collection of genomes. The authors successfully date the acquisition of *qacA* in a specific methicillin-resistant ST45 lineage in Australia and Singapore and find that the gene is located chromosomally, in a variable region alongside other crucial antimicrobial resistance genes. Additionally, the article presents evidence supporting the possible role of *qacA* in heightened tolerance to chlorhexidine, backed by mutagenesis and phenotypic assays. Overall, the manuscript is interesting, well-written, easy to follow and experimentally sound.

I only have a few comments to make:

Reviewer Comments:

1. In the introduction the authors justify the relevance of their study in investigating the possible community circulation of *qacA*-carrying MRSA. However, there is no information regarding the origin (hospital-acquired or community-acquired) of genomes in the results and discussion. If not available I would encourage the authors to mention this as a limitation. Also, the inclusion of SCCmec typing data would be a nice addition to the manuscript that could shed some light regarding the circulation of community-associated strains (SCCmec type IV). This information would deepen the understanding of *qacA* acquisition in the MRSA ST45 lineage and its implications for community-associated infections.
2. The authors present a Bayesian analysis grouping genomes into seven clades based on genomic relatedness, geographic origin, and the presence/absence of *qacA* or other antimicrobial resistance genes (Figure 2A). While this approach appears well-founded, it is essential to address the absence of a third subclade within clade 2, considering the evident differences between the genomes from Australia and Singapore. By not creating a separate subclade, the authors risk obscuring potentially meaningful results in their population trend analyses. Indeed, the minor peak observed in clade 2B's trend between 2010 and 2015 could be influenced by the collection dates of Singaporean genomes in this subclade.
3. The authors have assembled a robust cohort of ST45 genomes to explore global evolution (n=2001). However, approximately 50% of these genomes lack country of origin information, which could impact the conclusions about the predominance of ST45 lineages in specific geographic areas. To address this, the authors should carefully consider the implications of incomplete country information on their analysis and findings. Additionally, adding a legend to Figure 1 explaining the meaning of "missing data" or "no country information" would enhance clarity for readers.
4. The study reports differences in MICs between *qacA*-carrying and non-carrying strains. As broth microdilution results can be subject to interpretation errors of up to 1 double dilution, it is essential for the authors to clarify whether the reported MIC values are the mean of six replicates or if all six replicates yielded the same result. This information will enhance the reliability of their findings and allow

readers to better understand the precision and reproducibility of the MIC data.

5. Minor comments:

5.1 The inset of Figure 2B within Figure 2A lacks clarity, making it confusing for readers to distinguish each panel. Adding a frame around the inset would improve visual separation and facilitate better comprehension of the figure's content.

5.2 Considering the journal's scope and potential readership, the use of "delta qacA" instead of "qacA isogenic mutants" may be considered. This change aligns widely recognized nomenclature and enhances the accessibility of the article to a broader audience.

Reviewers' comments:

Reviewer #1 (Remarks to the Author):

The current study presents a bioinformatics analysis of a large collection of *S. aureus* ST45 and reports the presence of *qac* genes, which is known to confer resistance to biocides. Moreover, the authors present a timed phylogeny and are able to predict, with confidence, when the acquisition of the *qac* genes happened in ST45. The bioinformatics analyses are supplemented by *in vitro* experiments, in which authors demonstrate that indeed the presence of *qac* genes is responsible for decreased susceptibility to chlorhexidine, a biocide widely used in clinical settings.

The bioinformatics methods employed in the current study are sound and state of the art, however, the information presented in the manuscript is not novel, in the sense that presence of *qac* genes in MRSA and ST45 in particular has been widely reported before.

Response:

We thank the reviewer for the encouraging comments on our bioinformatic analyses and phenotypic work. We agree with the reviewer that the carriage of *qacA* is not a novel finding in *S. aureus*, as highlighted in the introduction of the original manuscript (Lines 60 - 63). Numerous population studies have shown that successful *S. aureus* lineages, particularly MRSA, disseminate and evolve via clonal expansion. However, the contributing factors to clonal expansion are varied, and importantly, have been shown to differ between *S. aureus* sequence types (STs). Subsequently, the findings of previous studies considering multiple sequence types other than ST45 or genomically undefined datasets, while important, cannot be broadly applied to all *S. aureus* lineages. Our study focuses specifically on the role of *qacA* in the successful clonal expansion of ST45 *S. aureus*, a pandemic lineage that is increasing in prevalence, and goes far beyond a simple measure of *qacA* gene prevalence in this lineage.

For example, an Irish study on samples collected between 2000-2012 <https://doi.org/10.1128/aac.02653-13> has reported *qac* genes in various *S. aureus* sequence types.

This Irish study examined 88 sporadic MRSA isolates collected over 12 years. The prevalence of *qacA* was reported but only in ST5, ST8, ST22, and ST239. This could be considered a secondary finding as there was no further mention of *qacA* in the results or in the discussion. Because of the small sample size used in this study

relative to its long sampling time frame, the authors did not conclude a clonal association with *qacA* carriage.

Another paper from 2011 (<https://doi.org/10.1128/aac.02653-13>) on the prevalence of *qac* genes in samples from nurses and general population in HK found that 50% of MRSA isolates carry *qac* genes.

The study in Hong Kong (<https://doi.org/10.1016/j.jhin.2011.02.018>) reported that *qacA* was detected in 50% (6/12) of MRSA, a very small sample size, and the authors did not state the significance of *qacA* positivity at the population level. No molecular typing and genomic characterisation of *S. aureus* were performed to investigate clonal associations. Further, the chlorhexidine tolerance phenotypes reported overlapped between the *qacA*-positive and *qacA*-negative isolates. This is in line with the knowledge gap highlighted in the introduction (Lines 70 – 75) and discussion (Lines 299 - 312) of our original manuscript, regarding the known issues with antibiotic susceptibility assays being used for assessing biocide tolerance and the importance of our mutagenesis work in the confirmation of *qacA*-mediated biocide tolerance in different *S. aureus* lineages.

A 3-year study on 878 isolates from Singapore (<https://doi.org/10.1016/j.cmi.2018.12.036>, not cited in the manuscript currently under review) found that "Among *qacA/B* carrying MRSA, the majority of *qacA/B* was found in ST45 (n = 287/409; 70.2%), followed by ST22 (n = 71/409; 17.3%), and other MRSA STs (n = 51/409; 12.5%)."

We thank the reviewer for highlighting the importance of this Singaporean study. In agreement with the reviewer, this study appears to be the first that reported a strong clonal association between ST45 MRSA and was included in our original manuscript (reference #31; reference #12 in the revised manuscript).

The authors of the Singaporean study stated that "Although almost all (99%) of our ST45 MRSA carried *qacA/B*, they were not positively associated with reduced susceptibility to chlorhexidine." and provided the rationale for the mutagenesis work and phenotypic characterisation of *qacA* in ST45 MRSA in our study.

Despite the high prevalence of *qacA*, the authors of the Singaporean study did not further investigate the structural genomics of *qacA* and how this correlated with population structure. Further, the genomic data generated in the Singaporean study has not been made publicly accessible. As such, we were unable to directly connect our findings to what was reported in this Singaporean study.

We believe that we have presented novel findings demonstrating the acquisition of a chromosomal *qacA* in conferring biocide tolerance in relation to the evolution of ST45 MRSA using phylogenetics, comparative genomics, population genetics, and mutagenesis, all of which were not employed in the three studies above.

It is my belief that the timed phylogeny and the results of the in vitro experiments presented in this manuscript do not add enough novel information for the manuscript to be published in the current journal.

To address the reviewer's comments, we have added the following sentences to emphasise the current knowledge gap and the novelty of our work.

- In Lines 70 – 72 of the revised manuscript, we added examples showing *qacA* carriage may not always associated with chlorhexidine tolerance. "In addition, discrepancies between *qacA* carriage and phenotypic tolerance to chlorhexidine (minimum inhibitory concentration (MIC) < 4 mg/L) have been reported at the population level."
- In Lines 99 – 102 of the revised manuscript, we highlighted the missing knowledge in the literature "Nonetheless, to our knowledge, the contribution of *qacA* to the expansion of ST45 MRSA in Australia has not been explored. Given the worldwide dissemination of ST45 *S. aureus*, this knowledge gap warrants the need to investigate the genomic correlation between *qacA* and the evolutionary dynamics of ST45 MRSA."

Reviewer #2 (Remarks to the Author):

The article by Nong and colleagues explores the dissemination of ST45 *Staphylococcus aureus* from a phylogenomic and phenotypic perspective, and attempt to understand the role of the acquisition of *qacA*, a resistance determinant involved in tolerance to biocides. The study utilizes a well-structured design and employs various techniques, including phylogenomic and population dynamic analyses, to investigate a robust collection of genomes. The authors successfully date the acquisition of *qacA* in a specific methicillin-resistant ST45 lineage in Australia and Singapore and find that the gene is located chromosomally, in a variable region alongside other crucial antimicrobial resistance genes. Additionally, the article presents evidence supporting the possible role of *qacA* in heightened tolerance to chlorhexidine, backed by mutagenesis and phenotypic assays. Overall, the manuscript is interesting, well-written, easy to follow and experimentally sound.

I only have a few comments to make:

1. In the introduction the authors justify the relevance of their study in investigating the possible community circulation of *qacA*-carrying MRSA. However, there is no information regarding the origin (hospital-acquired or community-acquired) of genomes in the results and discussion. If not available I would encourage the authors to mention this as a limitation. Also, the inclusion of SCCmec typing data would be a nice addition to the manuscript that could shed some light regarding the circulation of community-associated strains (SCCmec type IV). This information would deepen the understanding of *qacA* acquisition in the MRSA ST45 lineage and its implications for community-associated infections.

Response:

We thank the reviewer for highlighting this limitation of missing information and providing valuable insight into the importance of SCCmec typing in improving our understanding of ST45 genomic background in relation to *qacA* carriage.

To address this comment, we first added the following sentence in the introduction (Lines 89 – 91) to further emphasise the impact of ST45 MRSA in our setting. "Of particular note, the increased rate of multidrug resistance among community-associated MRSA (i.e. from 9.2% in 2013 to 13.7% in 2019) was primarily driven by ST45."

We do not have metadata regarding the hospital or community origin for the publicly available genomic sequences, but we have included the "hospital-associated" and "community-associated" status for the isolates from Australian surveillance programs where available in Table S2. In response to this comment, we added the following sentences in the revised manuscript.

- In Lines 170 – 172, we added "The available metadata on the hospital and community origin of these isolates was limited (Table S2). *qacA* was detected in 45.7% (16/35) of the community-associated isolates and 52% (13/25) of the hospital-associated isolates."
- In Lines 355 – 358, we added "Fourth, we did not possess metadata regarding the community or hospital origin for the majority of isolates included in the phylogenetic analyses (Table S2). This limitation limited our ability to estimate the burden of *qacA*-harbouring isolates and understand their adaption to settings with routine antiseptic use."

We have performed additional *in silico* SCCmec typing for all the isolates used in this study. This data has been added to Table S2. We have made the following changes to the manuscript:

- In Lines 116 – 118, we edited "The chromosome harboured *mecA* located in a staphylococcal cassette chromosome *mec* (SCCmec) type V element; biocide tolerance genes (*qacA*, *qacR*); and resistance genes to penicillins (*blaZ*, encoding PC1 β -lactamase), tetracyclines (*tetK*), macrolides (*ermC*), and aminoglycosides (*aac(6')-aph(2'')*)."
- In Lines 137 – 138, we added "Of the entire dataset, the vast majority of *qacA* harbouring isolates (97.4%, 411/422) belonged to SCCmec type V (Table S2)."
- In Lines 164 – 168, we added/edited "Of further note, SCCmec type differed between the *qacA*-negative (Type IV, 9/13) and *qacA*-positive isolates (Type V 4/13) in the C2A subclade (Table S2). Moreover, *qacA*-negative isolates formed two large C4 and C6 (Australian) subclades and a small C5 (American) subclade. The isolates in these *qacA*-negative subclades belonged to SCCmec type V (100%, 152/152) and lacked *aac(6')-aph(2'')* (0%, 0/152), but were frequently associated with *tetK* (83.4%, 126/152) and less often carried *ermC* (49.3%, 75/152)."
- In Lines 268, we edited "In agreement with a recent study examining the emergence of ST45 *S. aureus*, our analysis using a larger dataset revealed a clonally diverse global population structure with SCCmec Type V MRSA predominated among Australian isolates."
- In Lines 401 – 403, we added "SCCmec typing was performed using staphopia-sccmec v1.0.0, and SCCmecFinder v.1.2 was used to distinguish between SCCmec Type V and VII."

2. The authors present a Bayesian analysis grouping genomes into seven clades based on genomic relatedness, geographic origin, and the presence/absence of *qacA* or other antimicrobial resistance genes (Figure 2A). While this approach appears well-founded, it is essential to address the absence of a third subclade within clade 2, considering the evident differences between the genomes from Australia and Singapore. By not creating a separate subclade, the authors risk obscuring potentially meaningful results in their population trend analyses. Indeed, the minor peak observed in clade 2B's trend between 2010 and 2015 could be influenced by the collection dates of Singaporean genomes in this subclade.

Response:

We thank the reviewer for highlighting this limitation in our phylogenetic analysis. We agree that the presence of a third subclade comprising Singaporean isolates in Clade C2 of the maximum clade credibility tree (Figure 2A) may obscure the result interpretation.

We wanted to apply a consistent approach to defining subclades (Figure 2A), and this was based on temporal phylogeny, geographic origin, and the presence/absence of several key antimicrobial resistance determinants including *qacA*. While these Singaporean isolates appeared to be evolutionarily distinct from the Australian isolates in Clade C2, there was no clear difference in the antimicrobial resistance profile displayed in the heatmaps (Figure 2A). In comparison, the lack of key antimicrobial resistance determinants was highlighted in the Australian isolates collected in 2019, leading to the separation of Clade C2A for further investigation on gene loss. Further, these Clade C2 Singaporean isolates were collected in the same year and if treated as a separate clade would not have a temporal signal in the Bayesian phylogenetic modelling.

We do have unreported findings where we tested all Clade C2 isolates together in the in the GMRF Skyride reconstruction. However, the inclusion of C2A isolates (collected in 2019) caused a sudden collapse in the effective population size, indicating the model was not appropriate for analysing the C2 Clade as a whole.

To address this comment, we have made the following changes to the revised manuscript:

- To enhance the clarity of the subclade definition for Clade C2A, we edited the following sentences (Lines 151 – 154): “For population genetic and comparative genomic analyses, these isolates were divided into seven subclades, numbered C1 to C7, based on temporal phylogeny, geographic origin, and presence/absence of key antimicrobial resistance determinants including *qacA* (Figure 2A).” and (Lines 161 – 164) “However, a lack of these AMR determinants

including *qacA* was noted in the most recent Australian isolates (69.2%, 9/13) collected in 2019, resulting in the separation of C2A subclade for further investigation on gene loss.”

- We have also edited the following sentence (Lines 181 – 182) to highlight the effect of Singaporean isolates on the effective population size interpretation, “Of note was the rapid increase in the median EPS of C2B between 2008 and 2018 (Figure 2B), with the first peak observed in 2014 attributed to the Singaporean isolates.”
- As shown below, we have added a new supplementary figure, replacing Figure S1, to acknowledge the population collapse caused by Clade C2A isolates using the Skyride model. In the revised main manuscript (Lines 182 - 183), we have added the following sentence to describe this issue. “Extended analysis using the entire C2 subclade showed a population collapse in 2019 due to the inclusion of C2A subclade (Figure S1A).” The order of the supplementary figures has been changed accordingly in the main manuscript and the supplementary materials.

Figure S1 – Gaussian Markov random field Bayesian skyride plot displaying effective population size (EPS) of the C2 (dark green), C4 (navy blue), and C6 (light blue) subclades. The solid black line indicates the median EPS, and the coloured boundary indicates the 95% highest posterior density (HPD).

3. The authors have assembled a robust cohort of ST45 genomes to explore global evolution (n=2001). However, approximately 50% of these genomes lack country of origin information, which could impact the conclusions about the predominance of ST45 lineages in specific geographic areas. To address this, the authors should carefully consider the implications of incomplete country information on their analysis and findings. Additionally, adding a legend to Figure 1 explaining the meaning of "missing data" or "no country information" would enhance clarity for readers.

Response:

We thank the reviewer for highlighting this limitation. To address this comment, we have added the following sentence to the Figure 1 legend (Line 564 – 565), “Isolates with unknown geographic information were denoted by the blank heatmaps on the outer ring”.

To further discuss the limitation of unknown geographic information, we have made the following changes in the revised manuscript.

- In Lines 347 – 348, we edited “First, the missing geographic information for a large number of isolates included in the global phylogeny of ST45 *S. aureus* and the overrepresentation of Australian and Singaporean isolates sampled across limited time frames for Bayesian analysis should be noted, limiting our capacity to fully understand the global emergence and dissemination of this *qacA*-harbouring lineage.”
- In Lines 363 – 365, we added “(iv) enhancing Bayesian phylogenetics on investigating the origin and dissemination of the *qacA*-harbouring lineage/sublineages using additional historic and contemporary isolates with diverse geographic background.”

4. The study reports differences in MICs between *qacA*-carrying and non-carrying strains. As broth microdilution results can be subject to interpretation errors of up to 1 double dilution, it is essential for the authors to clarify whether the reported MIC values are the mean of six replicates or if all six replicates yielded the same result. This information will enhance the reliability of their findings and allow readers to better understand the precision and reproducibility of the MIC data.

Response:

We thank the reviewer for highlighting this detail regarding MIC interpretation. We are aware of the common interpretation errors with MIC testing and have added the following revisions to more clearly state the replicate testing that was performed. “The MIC and MBC results were consistent across biological replicates” (Lines 243 - 244 in the revised main manuscript). We have further clarified that only three biological replicates were performed for MIC and MBC testing (Line 486), and six biological replicates were performed for the growth assays (Line 487).

5. Minor comments:

5.1 The inset of Figure 2B within Figure 2A lacks clarity, making it confusing for readers to distinguish each panel. Adding a frame around the inset

would improve visual separation and facilitate better comprehension of the figure's content.

Response:

We thank the reviewer's comment on improving the clarity of the figures. In response, we have highlighted the frame of Figure 2B to allow better visual separation from Figure 2A as shown below.

5.2 Considering the journal's scope and potential readership, the use of "*ΔqacA*" instead of "*qacA* isogenic mutants" may be considered. This change aligns widely recognized nomenclature and enhances the accessibility of the article to a broader audience.

Response:

We agree with the reviewer. We have changed the nomenclature of "*qacA* isogenic mutant" and "*qacA* complemented strain" to "*ΔqacA* strain" and "*ΔqacA:qacA* strain", respectively, in the revised main manuscript (Lines 239, 243, 249 – 254, 257 – 259, 260, 318, 467, 477 – 478, 494 – 495, and 500), Figure 3 labels and figure legend (Lines 588 – 589, and 592), Figure S7 labels and figure legend (Lines 59 - 60, and 64) below, and Table S4.

Figure 3 - *In vitro* competition assays using co-cultures of AUSMDU00020487 wild-type paired with $\Delta qacA$ strain, and $\Delta qacA:qacA$ paired with $\Delta qacA$ strain. These co-cultures were exposed to either non-selective (purple) or selective condition using 0.5 mg/L CHG (yellow) for 7 days. Three biological replicates were performed for each condition tested. The mean percentages of wild-type or $\Delta qacA:qacA$ strains on Day 1 and Day 7 post-exposure are displayed, with the black error bars representing the standard error of the mean. Asterisks denote statistically significant differences as determined by paired *t* test (* $p \leq 0.05$ and *** $p \leq 0.001$).

Figure S7 – Doubling time of AUSMDU00020487 wild-type, $\Delta qacA$, and $\Delta qacA:qacA$ strains. Six biological replicates were used for determining doubling times depicted with Tukey whisker boxplots. The outlier is indicated by the grey dot. No significant difference (ns, $p > 0.05$ by unpaired t test) in the mean doubling time was observed among the wild-type (mean \pm standard error of the mean = 21.64 ± 0.27 min), $\Delta qacA$ (22.81 ± 0.72 min), and $\Delta qacA:qacA$ strain (22.59 ± 0.71 min).

REVIEWERS' COMMENTS:

Reviewer #3 (Remarks to the Author):

Nong and colleagues present a revised manuscript investigating the clonal expansion of ST45 *S. aureus* and the effects of *qacA* on biocide resistance.

The role of this reviewer was to assess the response of the authors to the previous reviewer comments.

This reviewer agrees with the previous reviewers. The authors use a wide collection of ST45 isolates. The performed analysis is sound and state of the art.

In response to the comments of reviewer 1, the manuscript refers now more appropriately to previous studies, highlights their limitations and it becomes more evident that the current datasets offer novel and deeper insight into the development of this important lineage.

The authors do also present convincing responses to comments of review 2. Strain information was updated, *in silico* SCCmec-typing was performed. New Supplementary figures are provided for the Bayesian analysis and MIC analysis is now appropriately described to allow interpretation.